# Causal inferences and real-world evidence: A comparative effectiveness evaluation of abiraterone acetate against enzalutamide

Per Johansson[1,2,3,4☯], Paulina Jonéus[1☯], Sophie Langenskiöld[2,5☯]*

1 Department of Statistics, Uppsala University, Uppsala, Sweden, 2 Centre for Health-Economic Research, Uppsala University, Uppsala, Sweden, 3 Institute for Evaluation of Labour Market and Education Policy, Uppsala University, Uppsala, Sweden, 4 Tsinghua University, Beijing, China, 5 Department of Medical Sciences, Uppsala University, Uppsala, Sweden

☯ These authors contributed equally to this work.
* Sophie.langenskiold@medsci.uu.se

## Abstract

Regulatory authorities are recognizing the need for real-world evidence (RWE) as a complement to randomized controlled trials in the approval of drugs. However, RWE needs to be fit for regulatory purposes. There is an ongoing discussion regarding whether pre-publication of a protocol on appropriate repositories, e.g. ClinicalTrials.gov, would increase the quality of RWE or not. This paper illustrates that an observational study based on a pre-published protocol can entail the same level of detail as a protocol for a randomized experiment. The strategy is exemplified by designing a comparative effectiveness evaluation of abiraterone acetate against enzalutamide in clinical practice. These two cancer drugs are prescribed to patients with advanced prostate cancer. Two complementary designs, including pre-analysis plans, were published before data on outcomes and proxy-outcomes were obtained. The underlying assumptions are assessed and both analyses show an increased mortality risk from being prescribed abiraterone acetate compared to enzalutamide.

## 1 Introduction

There is a growing interest in using real-world evidence (RWE) for regulatory purposes. The belief is that real-world data (RWD), or observational data, can make drug development more efficient and speed up patient access to new drugs. The European Medicines Agency (EMA) was therefore paving the way for RWE. EMA is providing incentives to use RWE for regulatory approval by, for example, the introduction of the Adaptive Pathways pilots in March 2014 [1]. The Adaptive pathways offered an iterative process for regulatory approval in which data from randomized control trials (RCT) are supplemented with RWD. Additionally, EMA is revising pharmaceutical legislation to acknowledge the possibilities arising from RWD analyses to support the development, authorization, and use of medicines [2] (cf. Burns et al. [3]).

However, there is concern that analyses based on observational data suffer from substantial biases [4]. Consequently, there are numerous initiatives for methodological improvements to, among others, control biases. One such initiative is IMI GetReal: a joint effort between EMA,

**Data Availability Statement:** According to Swedish legislation (https://etikprovningsmyndigheten.se/?s=vad+säger), data cannot be made available beyond what has been

approved by the ethical review board. Therefore, the data cannot be made publicly available. The ethical permission to replicate our research and also the data for the replication need, however, be requested by the researchers themselves; the permission at registrator@etikprovningen.se, and the data at registerservice@socialstyrelsen.se, uppdrag@scb.se, or npcr@npcr.se for the NBHW, Statistics Sweden, and NPCR respectively. The practical arrangements for accessing the data will, to some extent, depend on the location of the researcher.

**Funding:** This research was supported by the Dental and Pharmaceutical Benefit Agency (02823/2017). The funders had no role in study design, data collection and analysis, decision to publish, or preparation of the manuscript.

**Competing interests:** The authors have declared that no competing interests exist.

the industry, and the EU, that offers an exchange of insight and know-how in using RWD in drug development [5]. Similar initiatives have been set up in the US and Asia [6]. The results of these are more authorizations of drugs and extensions of indications based on RWE. For new products and extensions of indications submitted to the Agency in 2018 and 2019, Flynn et al. [7] find that 40% of the initial marketing authorization applications and 18% of applications for products currently on the market contained RWE.

As RWD is increasingly accepted as evidence in the regulatory process, there is an ongoing discussion about whether or not the requirement for generating this evidence should be the same as for RCT. One such requirement is that of pre-published protocols in an appropriate repository, such as *ClinicalTrials.gov*. However, there has yet to be consensus on the content requirement in such a pre-published protocol [3]. As a potential input to the discussion, this paper illustrates that an observational study based on a pre-published protocol can entail the same level of detail as a protocol for an RCT.

To this end, we present the first set of results from a methodological project funded by the Swedish Dental and Pharmaceutical Benefits Agency (TLV). TLV is responsible for determining which pharmaceutical products, care-related medical devices, and dental care procedures should be subsidized by the Swedish state. The objective of the project was to serve as a template for how to use the Swedish administrative population registries, in combination with quality registers, to conduct comparative effectiveness evaluations of interventions. The first part of the project consists of a comparative effectiveness evaluation of abiraterone acetate (AA) against enzalutamide (ENZ) in clinical practice. These two cancer drugs are given to patients who have advanced prostate cancer. The second part consists of a comparative effectiveness evaluation of these two drugs against standard of care.

In this paper, the results from the first part of the project are discussed. The designs were described in two pre-analysis plans [8, 9], both published before access to outcome data.

The main advantage with detailed protocol requirement is that it restricts the potential for p-hacking, forking etc., which is a problem with empirical research, see e.g. Amrhein et al. [10]; Wasserstein et al. [11]. Thus, one can argue that analyses based on detailed pre-published protocols increase the analyses' objectivity. An objection to publishing a detailed protocol is that it restricts the possibility of the researchers incorporating new information only available after having access to all data. With access to data, the researcher may observe irregularities, enabling them to find a suitable model that will increase both the validity and precision of the analysis. As we are prone to see patterns where there are none (i.e. apophenia), we believe this strategy of finding suitable models is risky and prone to providing invalid inferences. Furthermore, the fact that an analysis is based on a detailed protocol does not prohibit additional exploratory analyses using better-suited models to incorporate new information.

A high-quality study should be based on a carefully crafted design. This requires (i) an understanding of the assignment to treatment (i.e. an understanding of prescription practice in the applications), (ii) clear statements of the assumptions made, and (iii) details of how these assumptions should be assessed. A requirement of a detailed protocol, where these three steps are discussed, forces researchers to "think beforehand", which may increase quality by forcing the researcher to carefully think through the design, while at the same time, as a consequence of "tying oneself to the mast", provides an objective analysis. The readers themselves need to judge whether the illustration in this paper provides support for this claim.

## 1.1 Prostate cancer and novel hormone treatment

Prostate cancer (PC) is reported to be the most commonly diagnosed form of cancer in Sweden. In 2016, 10,473 patients were diagnosed, creating a total pool of 107,752 PC patients. It is

also the diagnosis with the largest number deaths among all main diagnoses of men in Sweden, and almost all deaths arise when patients have progressed to the advanced metastatic castrate-resistant prostate cancer (mCRPC) stage. Approximately 10–20 percent of patients with PC develop mCRPC within five years of follow-up after initial therapy [12–14].

Various treatment alternatives are available for patients with mCRPC. In the last two decades, chemotherapy and novel hormone treatment (NHT) medications (of which the first two in this group of treatments are AA and ENZ) have revolutionized treatment of mCRPC patients [15–20]. Patients with mCRPC have a poor prognosis, and their quality of life deteriorates as the disease progresses. When used for metastatic hormone-resistent prostate cancer, both AA and ENZ have thus shown to reduce mortality and improve overall survival [21].

Several indirect analyses have compared overall survival in patients treated with ENZ or AA, see, e.g. Chopra et al. [22], Fang et al. [23] or McCool et al. [24]. To the best of our knowledge, few studies have evaluated the comparative effectiveness of AA and ENZ on overall survival in a real-world setting. Recently, however, Tagawa et al. [25] and Schoen et al. [26] found an improved survival of ENZ over AA, using the Veterans Health Administration (VHA) database in the US.

This study evaluates the use of ENZ and AA in clinical practice from June 2015, corresponding to the date when these drugs were first reimbursed for mCRPC patients in Sweden. Data are collected from population registers administrated by the National Board of Health and Welfare (NBHW), Statistics Sweden (SCB), and the National Prostate Cancer Register (NPCR). The population is restricted to all men in the NBHW register with a prostate cancer diagnosis before 2017, as only these patients were expected to progress to mCRPC during the period for which we have outcome data. Before June 2015, almost no patients were prescribed any of the drugs as these were not yet reimbursed. The Dental and Pharmaceutical Benefit Agency (TLV) approved reimbursement for mCRPC patients who had failed androgen deprivation therapy (ADT) and were not yet suited for docetaxel (pre-chemotherapy), and for patients who had failed docetaxel (post-chemotherapy). Once reimbursed, the number of prescriptions increased rapidly. On 15 June 15 2018, AA was additionally reimbursed as an add-on to ADT in patients with high-risk metastatic hormone-sensitive prostate cancer (mHSPC). Therefore, we restricted the population to patients prescribed AA or ENZ from 1 June 2015, to 15 June 2018.

We estimate the effect on one primary outcome and two secondary outcomes to capture different aspects of morbidity. The primary outcome is all-cause mortality; the two secondary outcomes are skeleton-related events (SRE) and severe pain. The designs of the two complementary analyses are described in two pre-analysis plans [8, 9]. In order for the designs to be valid, there needs to be some randomness in the prescription given observed covariates. Section 2 provides arguments for why the doctors' prescription is random given the covariates used to control for the patients' health.

This study contributes to the growing literature of simultaneously using matching samples and instrumental variables analysis [27].

**Design 1** Johansson et al., [8] presents a matching design and protocol for a regression-adjusted matching estimator, where balance on observed covariates is obtained. Before publication, the design was discussed with two pharmaceutical companies, who had no objections and no requirement for additional covariates to be balanced.

**Design 2** Johansson et al., [9] is based on the same study population and makes use of differences in prescription practices across 21 county councils in an Instrumental Variables (IV) analysis.

The quality of healthcare, however, affects health, so quality differences across county councils can be related to the prescription of the two drugs. Thus, it is only relevant to believe the

instruments to be 'randomly assigned', given a set of controls for these quality differences. Johansson et al. [9] show the validity of the IV, present a sensitivity analysis of the exclusion restriction of the instruments, and pre-specify the IV models used for the analysis presented in this paper.

Inference from observational studies can potentially suffer from many types of biases, one of which concerns the potential objectivity of researchers. In this paper, access to outcome data was obtained after the publication of both pre-analysis plans (see S1 Text for documentation). The implication is that results from the analyses are as objective as those from an RCT. Furthermore, when adding mortality data, data from NPCR were also added. These data contain more detailed information on the patients' prostate cancer health, allowing us to assess the identifying assumptions in the designs and analyses.

The results from the two analyses show an increased mortality risk from prescribing AA compared to ENZ, and support the findings in Tagawa et al. [25] and Schoen et al. [26]. The matched sampling design analysis also suggests an increased risk of skeleton-related events. Further, the study shows the strength of using a matched sample design and IV strategy simultaneously. It also confirms previous results of lack of precision using the IV analysis [27].

The remainder of the paper continues as follows. Section 2 describes the data. Section 3 describes the matched sample design and the regression adjustment analysis, while section 4 describes the IV analysis. The results from both analyses are presented in section 5. In addition to the results on the mortality (section 5.1) and morbidity outcomes (section 5.2), this section assesses the potential problem of confounders (section 5.3) and a short discussion of the results from exploratory analyses (section 5.4). The paper concludes with a discussion in section 6.

## 2 Data

The specific data we use and the way we process the data have obtained ethical approval from the Ethical Committee in Uppsala (ref. Dnr2017/482). Data are collected only from population registers administered by the NBHW and SCB and quality registry administered by NPCR, which means that no informed consent was needed according to Swedish Act (2003:460) on Ethical Review of Research Involving Humans.

The data are then linked using unique serial numbers created by the SCB. From the NBHW, we link data from the inpatient care register and the pharmaceutical register. All inpatient and outpatient care visits in Sweden and all prescribed drugs are listed in these registers. The inpatient care register contains, among other things, information on all diagnoses (using the ICD classifications), date of admission, and discharge. The pharmaceutical register includes the date of prescribing and dispensing the drugs and the ATC class of the drug.

From SCB, we link data from a census conducted every fifth year over the period 1960–1990, labour statistics based on administrative sources (RAMS) for the period 1985–2009, and data from LISA covering the period 1990–2015. LISA is an extensive database that links a large set of administrative registers using the Swedish person id in the linkage. The linked data contain each individual's disposable income, labour income, social insurance payments, capital income, labour market status, year of birth, education, marital status, etc., from 1960 to 2015.

The population under study is defined using the cancer register. We first identify all men with a prostate cancer diagnosis before 2017 and the year of their diagnosis. We identify 243,535 unique patients with prostate cancer (ICD-10 code C61.9 or earlier codes ICD-7 177 and ICD-9 185.9). The population is then restricted to all men collecting a prescription of AA or ENZ from 1 June 2015 to 15 June 2018. The reasons for the time restriction are: (i) that almost no one was treated with these drugs before the reimbursement of AA and ENZ in June and July 2015, respectively, and (ii) that AA was additionally reimbursed in combination with

ADT in patients with high-risk castration hormone sensitive prostate cancer (mHSPC) and unsuited for docetaxel on 15 June 2018.

485 patients, or around 10%, of the patients were prescribed both AA and ENZ over the years. We allocate these patients to the two samples, AA and ENZ-takers, based on their first prescription of one of the two drugs (intention to treat analyses). The restriction leaves us with a total of 4,601 patients in the study population. The reason for this choice is that the first treatment can be considered as 'randomized' in the design, while the second cannot. However, we also present results from a sensitivity analysis in which we have excluded the 485 patients who were both prescribed AA and ENZ.

For this population, the year of the cancer diagnosis ranges between 1986 and 2016. Consequently, there is substantial variation in the time to be prescribed AA or ENZ from the date of cancer diagnosis. This so-called waiting time is most likely an important covariate.

As seen from Table 1, the prescription of the two drugs varies over the 21 county councils, hereafter denoted counties, the responsible body for healthcare in Sweden. The fact that the prescription varies substantially over counties is a notable finding as it suggests differences in the prescription that may not be related only to patients' health status. From this table, we can see that in total, 24 percent of the patients were prescribed AA, but the proportion ranges from 8 percent in Skåne to 61 percent in Kronoberg.

We have tried to understand the reason for this considerable regional variation by interviewing officials at TLV and doctors. The officials stated that it might be due to the negotiated price agreements between the county councils, the drug company and the Pharmaceutical Benefits Board. Confronted with this proposition, the doctors agreed; however, they stated

**Table 1. Proportion of patients prescribed ENZ and AA respectively, per county, year.**

| | ENZ | | | | AA | | | | ENZ | AA |
|---|---|---|---|---|---|---|---|---|---|---|
| | 2015 | 2016 | 2017 | 2018 | 2015 | 2016 | 2017 | 2018 | total | total |
| Blekinge | 0.84 | 0.91 | 0.84 | 0.80 | 0.16 | 0.09 | 0.16 | 0.20 | 0.85 | 0.15 |
| Dalarna | 0.86 | 0.65 | 0.55 | 0.62 | 0.14 | 0.35 | 0.45 | 0.38 | 0.69 | 0.31 |
| Gävleborg | 0.60 | 0.92 | 0.93 | 0.96 | 0.40 | 0.08 | 0.07 | 0.04 | 0.85 | 0.15 |
| Gotland | 0.75 | 0.92 | 1.00 | 1.00 | 0.25 | 0.08 | | | 0.89 | 0.11 |
| Halland | 0.75 | 0.89 | 0.88 | 0.76 | 0.25 | 0.11 | 0.12 | 0.24 | 0.82 | 0.18 |
| Jämtland | 0.47 | 0.81 | 0.77 | 0.67 | 0.53 | 0.19 | 0.23 | 0.33 | 0.70 | 0.30 |
| Jönköping | 0.62 | 0.91 | 0.89 | 0.76 | 0.38 | 0.09 | 0.11 | 0.24 | 0.80 | 0.20 |
| Kalmar | 0.74 | 0.89 | 0.83 | 0.93 | 0.26 | 0.11 | 0.17 | 0.07 | 0.84 | 0.16 |
| Kronoberg | 0.48 | 0.43 | 0.21 | 0.20 | 0.52 | 0.57 | 0.79 | 0.80 | 0.37 | 0.63 |
| Norrbotten | 0.83 | 0.88 | 0.64 | 0.65 | 0.17 | 0.12 | 0.36 | 0.35 | 0.74 | 0.26 |
| Örebro | 0.77 | 0.95 | 0.68 | 0.38 | 0.23 | 0.05 | 0.32 | 0.62 | 0.74 | 0.26 |
| Östergötland | 0.12 | 0.90 | 0.88 | 0.90 | 0.88 | 0.10 | 0.12 | 0.10 | 0.83 | 0.17 |
| Skåne | 0.87 | 0.96 | 0.96 | 0.85 | 0.13 | 0.04 | 0.04 | 0.15 | 0.92 | 0.08 |
| Södermanland | 0.77 | 0.98 | 1.00 | 0.92 | 0.23 | 0.02 | | 0.08 | 0.91 | 0.09 |
| Stockholm | 0.71 | 0.82 | 0.80 | 0.79 | 0.29 | 0.18 | 0.20 | 0.21 | 0.78 | 0.22 |
| Uppsala | 0.62 | 0.56 | 0.25 | 0.36 | 0.38 | 0.44 | 0.75 | 0.64 | 0.46 | 0.54 |
| Värmland | 0.87 | 0.96 | 0.67 | 0.75 | 0.13 | 0.04 | 0.33 | 0.25 | 0.83 | 0.17 |
| Västerbotten | 0.63 | 0.70 | 0.62 | 0.76 | 0.37 | 0.30 | 0.38 | 0.24 | 0.67 | 0.33 |
| Västernorrland | 0.47 | 0.71 | 0.73 | 0.86 | 0.53 | 0.29 | 0.27 | 0.14 | 0.64 | 0.36 |
| Västmanland | 0.73 | 0.79 | 0.80 | 0.55 | 0.27 | 0.21 | 0.20 | 0.45 | 0.74 | 0.26 |
| Västra-Götaland | 0.51 | 0.78 | 0.69 | 0.65 | 0.49 | 0.22 | 0.31 | 0.35 | 0.66 | 0.34 |
| Total | 0.68 | 0.83 | 0.76 | 0.74 | 0.32 | 0.17 | 0.24 | 0.26 | 0.76 | 0.24 |

that it also could be due to differences in hospital recommendations or habits. Unfortunately, the agreements are confidential, which means that we cannot provide evidence on price variation. However, we did se variation in recommendation across the county councils.

We construct 23 continuous covariates measuring a patient's general health and health progression before diagnosis and between diagnosis and treatment, including the number of visits at different periods and the number of days in inpatient care. The inclusion of the Elixhouser comorbidity index at diagnosis also captures the general health status.

Further, we include covariates separately for diseases deemed most important for prescription: cardiovascular diseases, metastases, diabetes, fatigue, and osteoporosis (see S1 Table for the included ICD codes). This results in 16 continuous covariates on the number of visits and eight indicators on whether or not a patient has had the specific diagnosis. We also derive three covariates measuring the number of collected prescriptions on medications, three years before the treatment, related to cardiovascular diseases and diabetes.

We create 91 variables intended to describe the socioeconomic status of the patient three years before diagnosis and three years before treatment, respectively, with SCB data from 1991 until 2015. For a few patients with diagnoses before or after these years, information on socioeconomic status is given by values from a year as close to the diagnosis as possible. These variables include information on age, marital status, educational level, pensions, income, sick leave, and other security benefits for the patient and their household. The mean over the three preceding years is used in the few cases of partly missing values on continuous covariates.

Educational level is the highest completed education and is classified as less than, equal to, or more than secondary school. There were 33 observations where education was reported as being unknown. Here a five-nearest neighbour approach is used to impute the missing values. The most common value of the five patients, i.e. neighbours, who are most similar in income, pension, age, and country of birth, is imputed for every missing value of the categorical variable measuring educational level.

One potential problem is that our data do not observe whether patients have received chemotherapy. Our inclusion of the time between diagnosis and treatment as a covariate is intended to control for this fact. In addition, as the quality of healthcare affects health, and may be related to the prescription of the two drugs, we also include the historical county-specific mortality related to prostate cancer at the year of diagnosis.

All 144 covariates with descriptions are presented in S2 Table. In the spring of 2018, 12 urologists and oncologists were asked about their prescription practices. Table 2 provides summary statistics of the AA and ENZ patients for a subset of the 15 variables judged by them to be the most important for the differences in prescription of the two drugs. In addition, the table also includes county-specific mortality rates and years to treatment.

From this table, we can see that the two groups are very similar. There are no significant differences in average age, educational level, or marital status. The main differences between the groups is that the ENZ patients have: (i) a higher prevalence of acute myocardial infarction, (ii) a higher prevalence of diabetes prescriptions, (iii) a lower prevalence of metastases, and (iv) a shorter time in years to treatment from diagnosis.

## 2.1 Outcome data

We have one primary and two secondary outcomes to capture different aspects of morbidity. The primary outcome is all-cause mortality; the two secondary outcomes are skeleton-related events (SRE) and severe pain.

All-cause mortality is defined using an indicator variable 'DEAD', taking value one for dead patients and zero for patients who are alive at the end of each 30-day period after beginning

**Table 2. Summary statistics.**

| Description | ENZ | AA | Diff. |
|---|---|---|---|
| Age at treatment | 75.27(7.85) | 75.29(7.70) | -0.02 |
| Years to treatment from diagnosis | 6.95(5.00) | 7.29(5.31) | -0.34* |
| Less than secondary school education | 0.36(0.48) | 0.34(0.48) | 0.02 |
| Secondary school education | 0.39(0.49) | 0.39(0.49) | 0.00 |
| Living with a partner | 0.66(0.47) | 0.67(0.47) | -0.01 |
| Country specific mortality at diagnosis, deaths per 1,000 inhabitants | 0.05(0.01) | 0.05(0.01) | 0.00 |
| Drugs used in diabetes, ATC A10, 3 years before treatment | 0.16(0.37) | 0.12(0.33) | 0.04*** |
| Beta blocking agents, ATC C07, 3 years before treatment | 0.38(0.49) | 0.37(0.48) | 0.01 |
| Calcium channel blockers, ATC C08, 3 years before treatment | 0.31(0.46) | 0.32(0.47) | -0.01 |
| Elixhouser score at diagnosis = 1–4 | 0.39(0.49) | 0.37(0.48) | 0.02 |
| Elixhouser score at diagnosis $>= 5$ | 0.02(0.15) | 0.02(0.15) | 0.00 |
| Osteoporosis before treatment | 0.01(0.11) | 0.01(0.10) | 0.00 |
| Metastases before treatment | 0.70(0.46) | 0.75(0.43) | -0.05*** |
| Acute myocardial infarction before treatment (I21) | 0.10(0.30) | 0.07(0.25) | 0.03*** |
| Atrial fibrillation and flutter before treatment (I48) | 0.16(0.36) | 0.18(0.38) | -0.02 |
| Other cardiovascular diseases before treatment | 0.27(0.45) | 0.26(0.44) | 0.02 |
| Fatigue before treatment | 0.05(0.22) | 0.04(0.20) | 0.01 |

Means, standard deviations within parentheses.

*p<0.1

**p<0.05

***p<0.01.

AA or ENZ treatment. Patients are assumed to suffer an SRE if they experience hospitalization because of pathologic fracture (ATC codes M485, M495, M844, and M907) or spinal cord compression (G550, G834, G952, G958, G959, and G992) [28]. The SRE indicator is valued as one for periods with such hospitalizations and zero for other periods. Patients are assumed to suffer severe pain if they receive prescriptions for neuropathic pain, i.e. opiates in combination with tramadol and paracetamol (ATC-codes N02AA, N02AX02, and N02BE01). The 'Pain' indicator is valued as one for periods in which the patient has received such a prescription and zero for other periods. With one primary and two secondary outcomes, we will use the Bonferroni corrected standard errors with a five percent overall level. This means that the level on the single outcomes will be 1.67% (= $100 \times 0.05/3$).

Of the 4,601 patients, 3,658 have a date of death (80 percent). Among those who died, the mean time between prescription and death was about 20 months. Of the 3,658 patients, 3,104 have prostate cancer as one of multiple causes of death, and 2,850 patients have prostate cancer as their main cause of death. Only 502 unique patients have an indication of SRE, and only 13 unique patients have an indication of pain.

## 3 Matched sample analysis

The matched sample was created using the generalized Mahalanobis distance metric, denoting genetic matching [29], with the objective of estimating an average treatment effect. As the two groups were not completely balanced, a one-to-one matching with replacement from a genetic matching algorithm was used.

Observations with a distance of more than three standard deviations for any of the included covariates were excluded. This led to 85 dropped observations and 4,516 observations in the

matched sample. Of these, 1,110 patients were prescribed AA, and the remaining 3,406 ENZ. The absolute mean differences in baseline covariates and after matching are shown visually in Fig 1. For details on the matching algorithm and the procedure of determining a sufficient balance of the covariates in the two groups, see Johansson et al. [8].

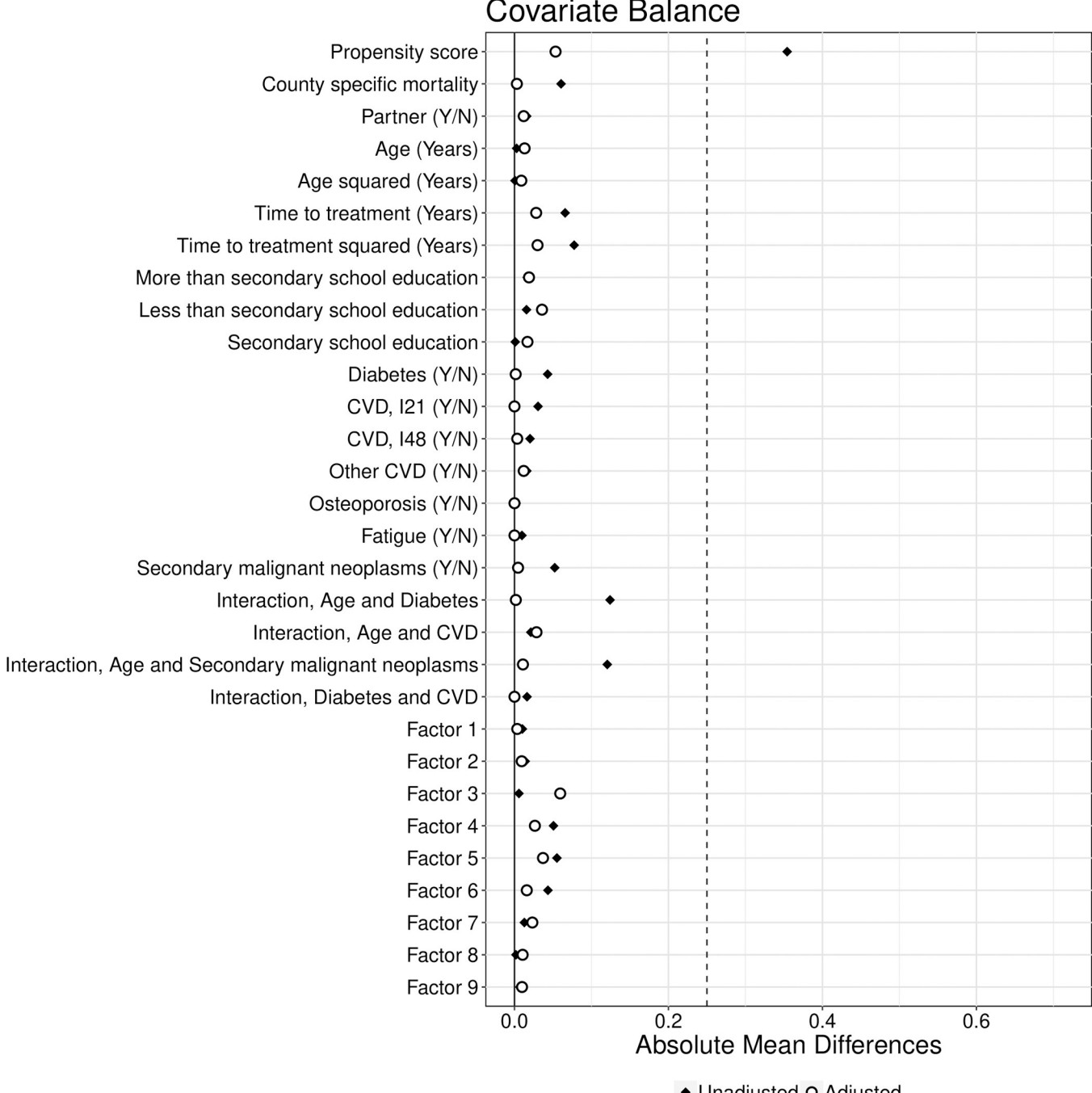

**Fig 1. Balance of the main covariates in the design.** Propensity score is estimated using a logit model, and using LASSO regression on all possible covariates. The factors are derived from an exploratory factor analysis on 130 continuous covariates. Factor 1: Welfare and social security benefits; Factor 2: Wages and disposable income; Factor 3: Occupational pensions; Factor 4: Early retirement benefits and welfare; Factor 5: Sickness and unemployment benefits; Factor 6: Private pensions; Factor 7: Health before treatment; Factor 8: Income at diagnosis; and Factor 9: Income from business.

## 3.1 The regression estimator

Taking stock of the Neyman-Rubin potential outcomes framework [30–32], we define the potential outcome if a patient had been given AA as $Y(1)$, and as $Y(0)$ if he had been given ENZ. Our interest is that of estimating the conditional average treatment effect for the population of $n$ individuals in our sample, formally defined as

$$CATE = \frac{1}{n}\sum_{i=1}^{n} Y_i(1) - Y_i(0) \tag{1}$$

Let $x_i$ be the observed covariates and let $W_i = 1$ if a patient was prescribed AA and $W_i = 0$ if prescribed ENZ. Under the Stable Unit Treatment Value Assumption (SUTVA,33), the observed outcome, $Y_i$, is equal to the potential outcome, thus $Y_i = Y_i(W_i)$.

The analysis sample is formed by finding the closest ENZ patient to an AA patient, and vice versa, regarding their covariates. In this procedure, we are matching with replacement. Formally, the unobserved outcomes $Y_i(0)$ $i = 1,\ldots,n_1$ for the AA patients with covariates $x_i$, are imputed as follows

$$\hat{Y}_i(0) = Y_{ij}, \quad i_j = \operatorname*{argmin}_{j=1,\ldots,n_0} \|x_j - x_i\|,$$

where $\|\cdot\|$ is the generalized Mahalanobis distance used in the genetic matching algorithm.

The unobserved outcomes $Y_j(1)$, $j = 1,\ldots,n_0$, for the ENZ patients, with covariates $x_j$, are then imputed as

$$\hat{Y}_j(1) = Y_{j_i}, \quad j_i = \operatorname*{argmin}_{i=1,\ldots,n_1} \|x_i - x_j\| \tag{2}$$

The one-to-one matching estimator of CATE is then defined as

$$\hat{\tau} = \frac{1}{n_1}\sum_{i:W_i=1}^{n_1}(Y_i - \hat{Y}_i(0)) + \frac{1}{n_0}\sum_{j:W_j=0}^{n_0}(\hat{Y}_j(1) - Y_j) \tag{3}$$

As seen from Eq (3), in the estimation we use twice as many observations as number of patients in the data. This means that observations are correlated and that this correlation needs to be considered in the inference. This is easily managed using a regression estimator in which the standard errors are to be estimated by clustering on individuals. An advantage of the regression estimator is that we can adjust for the bias due to inexact matching by adding covariates. Fig 1 shows the covariates used in the analysis.

We use the OLS estimator, and estimate the following regression model:

$$\tilde{Y}_h = \beta_0 + \tau W_h + x_h'\beta_1 + W_h \times (x_h - \bar{x})'\delta + \varepsilon_h, h = 1,\ldots,2n \tag{4}$$

where $\bar{x}$ is the sample mean of the covariates, $n = n_0 + n_1$ and $\tilde{Y}_h$ defines either the observed outcome or the imputed outcome. The OLS estimate, $\hat{\tau}$, is the estimate of the CATE. The standard errors are estimated by clustering at the individual level.

To obtain a summary measurement for the mortality outcome and to handle censoring, we also estimated a discrete time Cox regression model. That is, we let $\Pr(Y_{it} = 1|x_i, W_i, Y_{it-1} = 0) = \lambda_{it}(\theta)$, where

$$\lambda_{it}(\theta) = 1 - \exp(-e^{\gamma_t + W_i \times \tau + x_i\beta}), t = 1,\ldots,T-1,$$

where $T$ is the maximum number of months and $\theta = (\gamma_1,\ldots,\gamma_{T-1},\beta,\tau)$. Note that in this analysis we are using the 4,516 observations in the matched sample only.

Based on the maximum likelihood (ML) estimates (i.e. $\hat{\theta}$), we then estimate the survival function up to any month $t$ for the AA and ENZ patients, respectively. The estimated survival function up to $t$ is then

$$S(W, t|\hat{\theta}) = \frac{1}{n_{Wt}} \sum_{i:R_{Wt}} \prod_{\tau}^{t} (1 - \lambda_{it}(\hat{\theta})), \, W = 0, 1. \tag{5}$$

Here $R_{1t}$ and $R_{0t}$ are the risk set of the $n_{1t}$ and $n_{0t}$ patients that have not yet died in period $t$, respectively. The overall effect on survival up to a given period $\bar{T}$ is then estimated as

$$\hat{\Delta}_{\bar{T}} = \frac{1}{\bar{T}} \sum_{t=1}^{\bar{T}} S(1, t|\hat{\theta}) - S(0, t|\hat{\theta}) \tag{6}$$

## 4 Instrumental variable analysis

Almost always, researchers using IV estimators estimate the first-stage regression at the same time as conducting the analysis. Here, the first stage and the test for the relevance of the county instruments were made before observing our outcomes. This means that the following IV analysis is a design-based approach [cf. 33].

The framework for IV analysis is based on the model of potential latent variables. Under modelling assumptions, this allows deriving the IV estimator as a maximum likelihood (ML) estimator. This section describes this framework and the resulting ML estimator. Details regarding covariate definitions are provided in Johansson et al. [9].

The potential problem in identifying an average treatment effect in the matched data is that doctors could prescribe AA or ENZ based on health, which we cannot observe. To handle this potential problem, specify we specify a latent propensity for prescriptions. Let $q_{ic} = 1$ if individual $i$ is living in county $c$ and let $x_i$ be the covariates displayed in Fig 1 then this latent propensity for prescriptions is

$$W_i^* = \gamma'_x x_i + \gamma_c' q_i + \varepsilon_i, \tag{7}$$

where $q_i = (q_{i1,...,} q_{i21})'$ and $\gamma_c = (\gamma_{1,...,} \gamma_{21})'$. Let $\gamma = (\gamma'_x, \gamma'_C)'$, then $W_i^* = \gamma' z_i + \varepsilon_i$, where $z_i = (x'_i, q'_i)'$.

If $W_i^* > 0$, the patient is prescribed AA (i.e. $W_i = 1$), and ENZ otherwise. Under the further assumption that $\varepsilon_i$ is normally distributed, the probability of being prescribed AA is

$$\Pr(W_i = 1|z_i, \gamma) = \Pr(\varepsilon_i > -\gamma' z_i) = \Phi(\gamma' z_i), \tag{8}$$

where $\Phi(.)$ is the cumulative distribution function of the standard normal. This first stage regression was presented in Johansson et al. [9]. We tested for the relevance of $q_i$ (i.e. that $q_i$ affects prescription given the covariates), and assessed the exclusion restriction (i.e. no effect of $q_i$ on the outcome, except through $W_i$) by estimating the effect on Pain or SRE at the time of diagnosis as proxy outcomes. For completeness, we present the estimates from the probit estimation of (8), the test of relevance, and the assessment of the exclusion restriction in S2 Text.

Let the unobserved health of individual $i$ at month $t>0$ if given ENZ or AA be

$$Y_{it}(0)^* = \beta_{0t} + \delta'_t x_i + u_{i0t}$$

and

$$Y_{it}(1)^* = \beta_{1t} + \delta'_t x_i + \delta'_{Dt}(x_i - \bar{x}) + u_{i1t},$$

respectively. $\bar{x}$ is the mean vector of the covariates in the sample and $u_{i0t}$ and $u_{i1t}$ are error terms.

The unobserved health given the prescribed drug can then be formulated as a function of the error terms and the observed covariates:

$$Y_{it}^* = \beta_{0t} + \delta_t' x_i + \delta_{1t}' W_i + \delta_{Dt}' W_i(x_i - \bar{x}) + u_{i0t} + W_i(u_{i1t} - u_{i0t}). \qquad (9)$$

With this specification, $\delta_{1t}' = \beta_{1t} - \beta_{0t}$ is the average treatment effect at months $t$ on the latent outcome and vector $\delta_{Dt}'$ are heterogeneous effects, centred around $\delta_{1t}'$, with respect to the covariates.

The potential problem in the matched design is now solved in this model by letting $u_{i0t} = \rho_{0t}\varepsilon_i + \eta_{i0t}$ and $u_{i1t} = \rho_{1t}\varepsilon_i + \eta_{i1t}$, where $\eta_{0t}$ and $\eta_{1t}$ are both random. This means that the unobserved health is given as

$$Y_{it}^* = \beta_{0t} + \delta_t' x_i + \delta_{1t}' W_i + \delta_{Dt}' W_i(x_i - \bar{x}) + \rho_{0t}\varepsilon_i + \eta_{i0t} + (\rho_{1t} - \rho_{0t})W_i\varepsilon_i + W_i(\eta_{i1t} - \eta_{i0t}) (10)$$

Note that with this specification, we assume that there is no unobserved heterogeneity in the effects correlated with the county factor IV. This means that under model assumptions, we identify the average treatment effect instead of the complier treatment effect.

We observe $Y_{it} = 1$ if $Y_{it}^* \geq 0$ and $Y_{it} = 0$ if $Y_{it}^* < 0$. Under the assumptions that $\varepsilon_i$ is independent of $z_i$ (i.e. the exclusion restriction) and that $\varepsilon_i$, $\eta_{i1t}$ and $\eta_{i0t}$ are standard normal we get

$$\Pr(Y_{it} = 1 | z_i, W_i = 1, \varepsilon_i) = \Phi\left(\frac{\beta_{0t} + \delta_t' x_i + \delta_{1t} + \delta_{Dt}'(x_i - \bar{x}) + \rho_{1t}\varepsilon_i}{(1 - \rho_{1t}^2)^{1/2}}\right)$$

and

$$\Pr(Y_{it} = 1 | z_i, W_i = 0, \varepsilon_i) = \Phi\left(\frac{\beta_{0t} + \delta_t' x_i + \rho_{0t}\varepsilon_i}{(1 - \rho_{0t}^2)^{1/2}}\right)$$

As

$$\Pr(\varepsilon_i | \varepsilon_i > -\gamma' z_i) = \phi(\varepsilon_i)/\Phi(\gamma' z_i),$$

this means

$$\Pr(Y_{it} = 1 | z_i, W_i = 1) = \frac{1}{\Phi(\gamma' z_i)} \int_{-\gamma' z_i}^{\infty} \Phi\left(\frac{\beta_{0t} + \delta_t' x_i + \delta_{1t} + \delta_{Dt}'(x_i - \bar{x}) + \rho_{1t}\varepsilon_i}{(1 - \rho_{1t}^2)^{1/2}}\right)\phi(\varepsilon_i)d\varepsilon_i$$

where $\varepsilon_i$ in the integral is a dummy argument of integration. Similarly

$$\Pr(Y_{it} = 1 | z_i, W_i = 0) = \frac{1}{1 - \Phi(\gamma' z_i)} \int_{-\infty}^{\gamma' z_i} \Phi\left(\frac{\beta_{0t} + \delta_t' x_i + \rho_{0t}\varepsilon_i}{(1 - \rho_{0t}^2)^{1/2}}\right)\phi(\varepsilon_i)d\varepsilon_i$$

Let $p_i(\theta_{1t}) = \Pr(Y_{it} = 1 | z_i, W_i = 1)$ and $p_i(\theta_{0t}) = \Pr(Y_{it} = 1 | z_i, W_i = 0)$, where $\theta_{1t} = (\beta_{0t}, \delta_t', \delta_{1t}, \delta_{Dt}', \gamma', \rho_{1t})\prime$ and $\theta_{0t} = (\beta_{0t}, \delta_t', \gamma', \rho_{0t})\prime$, respectively. The likelihood to be

maximized with respect to $\boldsymbol{\theta}_t = (\beta_{0t}, \boldsymbol{\delta}'_t, \delta_{1t}, \boldsymbol{\delta}'_{Dt}, \boldsymbol{\gamma}', \rho_{1t}, \rho_{0t})\prime$ is then given as

$$
\begin{aligned}
L(\boldsymbol{\theta}_t; \boldsymbol{z}_i, Y_{it}, W_i) \\
= \prod_{i=1}^{n} \left[ p_i(\boldsymbol{\theta}_{1t})^{Y_{it}} (1 - p_i(\boldsymbol{\theta}_{1t}))^{(1-Y_{it})} \Phi(\boldsymbol{\gamma}'\boldsymbol{z}_i) \right]^{W_i} \\
\times \left[ p_i(\boldsymbol{\theta}_{0t})^{Y_{it}} (1 - p_i(\boldsymbol{\theta}_{0t}))^{(1-Y_{it})} (1 - \Phi(\boldsymbol{\gamma}'\boldsymbol{z}_i)) \right]^{(1-W_i)}.
\end{aligned}
$$

For each of the periods $t = 1,\ldots T$, we estimate the conditional individual treatment effect

$$
\hat{\Delta}(\boldsymbol{x}, t) = \Phi(\beta_{0t} + \hat{\boldsymbol{\delta}}'_t \boldsymbol{x}_i + \hat{\delta}_{1t} + \hat{\boldsymbol{\delta}}'_{Dt}(x_i - \bar{x})) - \Phi(\beta_{0t} + \hat{\boldsymbol{\delta}}'_t \boldsymbol{x}_i), \tag{11}
$$

where ^ denotes the maximum likelihood estimates. The conditional average treatment effect at each month $t$ is estimated as

$$
\hat{\Delta}_t = \frac{1}{n} \sum_{i=1}^{n} \hat{\Delta}(\boldsymbol{x}_i, t). \tag{12}
$$

This framework is easily extended to a discrete-time survival analysis model for the mortality outcome. Now we let $\lambda_i(\boldsymbol{\theta}_{1t}) = \Pr(Y_{it} = 1 | \boldsymbol{z}_i, W_i = 1, Y_{it-1} = 0)$ be the probability that patient $i$ prescribed AA dies at month $t$ given survival up to this month and let $\lambda_i(\boldsymbol{\theta}_{0t}) = \Pr(Y_{it} = 1 | \boldsymbol{z}_i, W_i = 0, Y_{it-1} = 0)$ be the corresponding conditional probability if the patient instead was prescribed ENZ. Furthermore, we let

$$
\lambda_i(\boldsymbol{\theta}_{1t}) = \frac{1}{\Phi(\boldsymbol{\gamma}'\boldsymbol{z}_i)} \int_{-\gamma' z_i}^{\infty} \Phi\left( \frac{\beta_t + \boldsymbol{\delta}'\boldsymbol{x}_i + \delta_1 + \boldsymbol{\delta}'_D(x_i - \bar{x}) + \rho_1 \varepsilon_i}{(1 - \rho_1^2)^{1/2}} \right) \phi(\varepsilon_i) d\varepsilon_i
$$

and

$$
\lambda_i(\boldsymbol{\theta}_{0t}) = \frac{1}{(1 - \Phi(\boldsymbol{\gamma}'\boldsymbol{z}_i))} \int_{-\infty}^{-\gamma' z_i} \Phi\left( \frac{\beta_t + \boldsymbol{\delta}'\boldsymbol{x}_i + \rho_0 \varepsilon_i}{(1 - \rho_0^2)^{\frac{1}{2}}} \right) \phi(\varepsilon_i) d\varepsilon_i.
$$

Thus, $\boldsymbol{\theta}_{1t} = (\beta_t, \boldsymbol{\delta}', \delta_1, \boldsymbol{\delta}'_D, \boldsymbol{\gamma}', \rho_{1t})\prime$ and $\boldsymbol{\theta}_{0t} = (\beta_t, \boldsymbol{\delta}', \boldsymbol{\gamma}', \rho_{0t})\prime$. The term $\beta_t + \boldsymbol{\delta}'\boldsymbol{x}_i$ measures the baseline conditional probability of dying, given survival up to month $t$, while $\delta_1$ and $\boldsymbol{\delta}'_D(x_i - \bar{x})$ measure the 'shift' in this baseline probability, i.e. an effect. We restrict the effect on the conditional probability to be the same at all months.

Let $\boldsymbol{\beta} = (\beta_1, \ldots, \beta_{T-1})\prime$, where $T$ is the last follow-up month and let $T_i$ be the number of the months the individual is alive, thus $T_i = T$, if the individual is alive when we end the study. The likelihood to be maximized with respect to $\boldsymbol{\theta}_t = (\boldsymbol{\beta}, \boldsymbol{\delta}', \delta_1, \boldsymbol{\delta}'_D, \boldsymbol{\gamma}', \rho_0, \rho_1)\prime$ is

$$
\begin{aligned}
L(\boldsymbol{\theta}_t; \boldsymbol{z}_i, Y_{it}, W_i) \\
= \prod_{i=1}^{n} \left[ p_i(\boldsymbol{\theta}_{1t})^{Y_{it}} (1 - p_i(\boldsymbol{\theta}_{1t}))^{(1-Y_{it})} \Phi(\boldsymbol{\gamma}'\boldsymbol{z}_i) \right]^{W_i} \\
\times \left[ p_i(\boldsymbol{\theta}_{0t})^{Y_{it}} (1 - p_i(\boldsymbol{\theta}_{0t}))^{(1-Y_{it})} (1 - \Phi(\boldsymbol{\gamma}'\boldsymbol{z}_i)) \right]^{(1-W_i)}.
\end{aligned}
$$

Based on the ML estimates, we then estimate the survival function up to any month t for the AA and ENZ patients (cf. Eq 5). The overall effect on survival up to a given period T̄ is estimated using Eq 6.

For the estimation of these models, we use the algorithm described in Huntington-Klein [34]. Confidence intervals of the estimand of interest are estimated using the bootstrap percentile method.

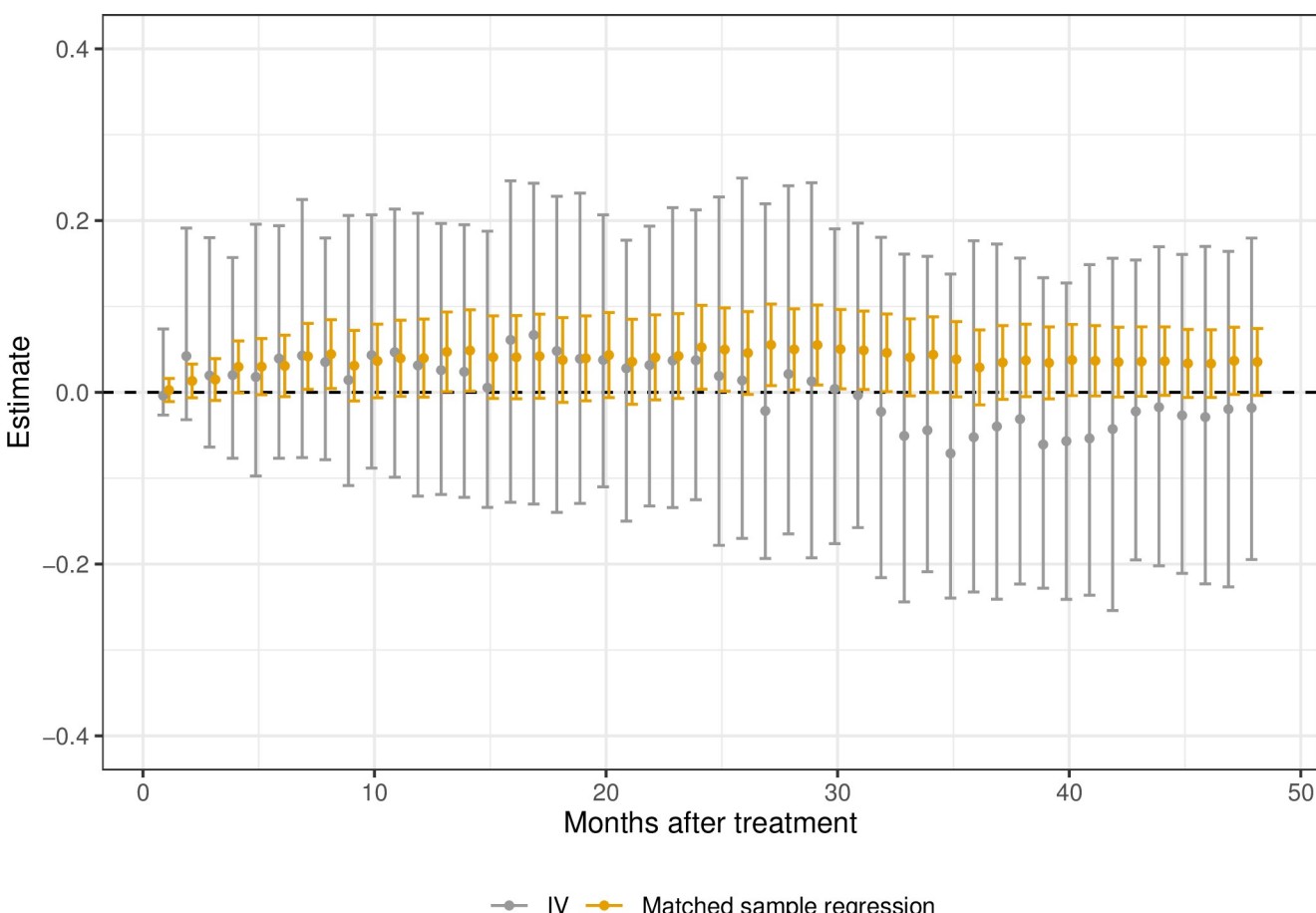

**Fig 2. CATE mortality, estimates (•) and 95% Bonferroni confidence intervals.** Bootstrap intervals for the IV, with the number of replicates, R = 500. All men dead before month $m$ have $Y_m = 1$.

## 5 Results

### 5.1 Mortality

The results for the matched sample and IV analysis are presented in Fig 2. The Fig displays the point estimate, and the 95% Bonferroni corrected confidence interval on 48 periods of 30 days, in the following denoted months. From the Fig, we can see 26 statistically significant effects, all from the matched sample regression. These results suggest a higher mortality rate from being prescribed AA than ENZ. The point estimates for the IV estimator are very similar. However, the confidence intervals are substantially wider (around four times as wide), explaining why no estimate is statistically significant.

The effects on survival time up to 48 months are, for both analyses, presented in Fig 3. Both show the same pattern: a clear reduction in survival rates for patients prescribed AA in contrast to ENZ. The overall effect on survival up to 48 months is estimated to $\hat{\Delta}_{48}^m = -0.38$ and $\hat{\Delta}_{48}^{IV} = -025$ in the matched sample and IV analysis, respectively. All of the estimates are statistically significant for the matched sample regression. As the length of the confidence intervals is around twice as long for the IV estimator as for the matched sample regression, the IV estimates are not statistically significant for short and long survival times.

**5.1.1 Sensitivity-analysis.** As discussed previously, we estimate the comparative effectiveness of participants treated as intended (intention-to-treat approach). Under the assumption

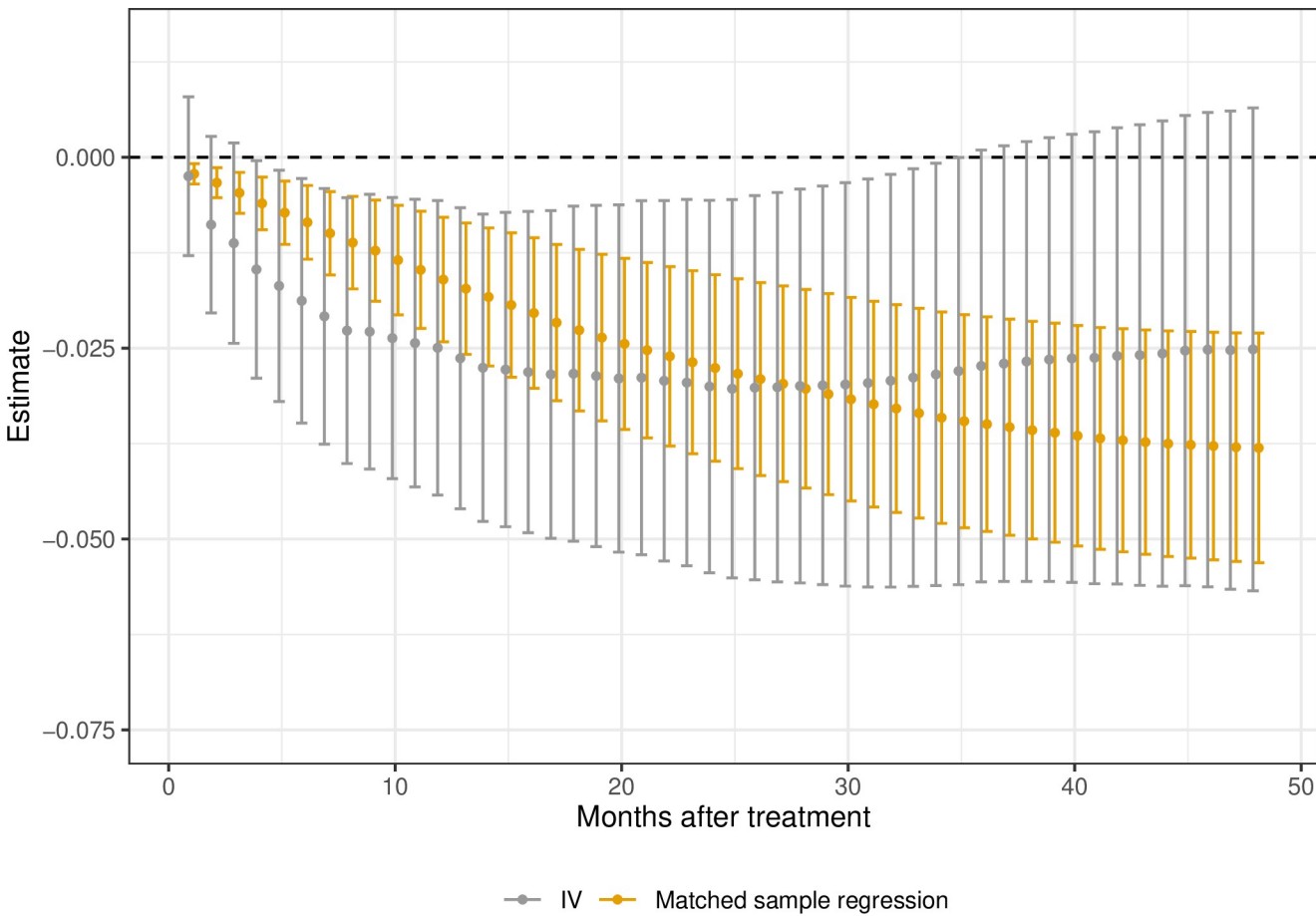

**Fig 3. The effect on survival time up to 48 months after treatment.** Estimates (•) and 95% Bonferroni confidence intervals. Bootstrap intervals with the number of replicates, R = 500.

that switching prescriptions among the NHTs is random, conditional on our observed covariates, we can also estimate the comparative effectiveness for participants following the protocol by excluding those who switch treatments from the analysis. This per-protocol analysis excludes 200 patients who were first prescribed AA and later prescribed ENZ, and 285 who made the reverse switch, i.e., among those 1,110 and 3,406 patients who were first prescribed AA and ENZ, respectively. This means that we exclude 18 percent and 8 percent of those first prescribed AA and ENZ.

The per-protocol analysis shows, again, a higher mortality rate from being prescribed AA than ENZ. The mortality differences in the matched analysis are more pronounced than in the intent-to-treat analysis, and the number of statistically significant effects is now 43 out of the 48 estimates (see S1 Fig). For the IV analysis the results are almost identical to the previous results and are therefore not included in the Supporting information. Consequently, under a stronger assumption, we can conclude that mortality is higher on AA than ENZ if the patients follow the protocol.

**5.1.2 Sub-analyses.** As detailed in Johansson et al. [8], three sub-analyses (responders, aggressiveness, and waiting time) on mortality were suggested for the matched sample analysis. Patients previously given hormone treatment for 12 months or more are defined as being high responders. Patients with visceral metastases in conjunction with hospitalization (ICD-10

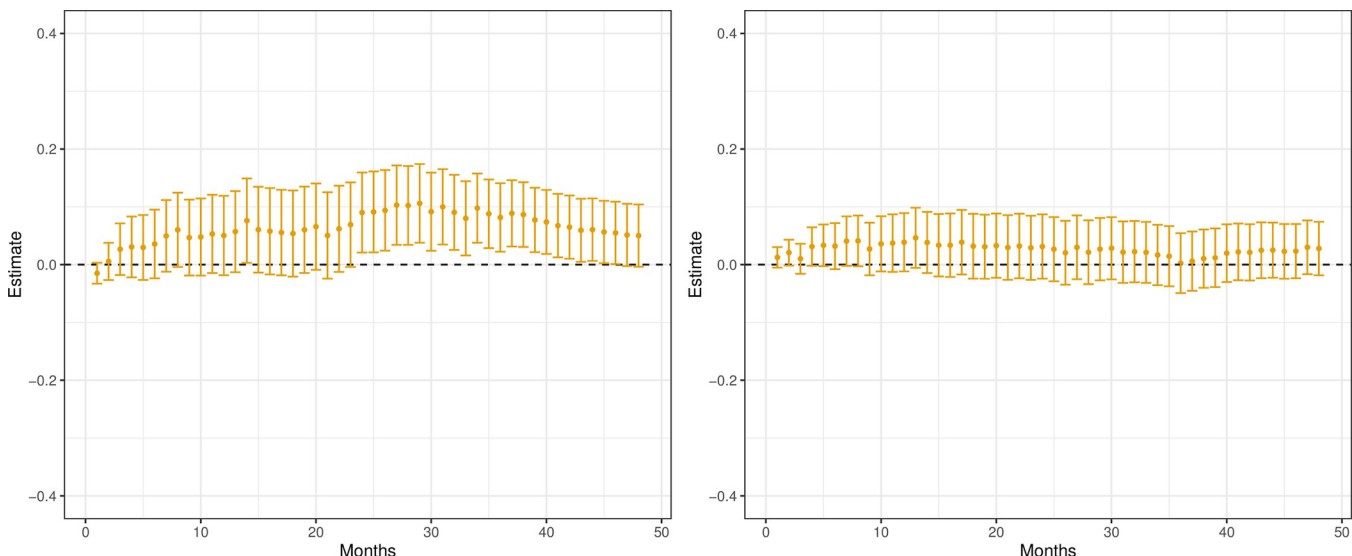

**Fig 4. CATE mortality, estimates (•) and 95% Bonferroni corrected confidence intervals.** Low and high responders. (a) Low respondents, i.e. patients previously given hormone treatment for less than 12 months. (b) High respondents, i.e. patients previously given hormone treatment for 12 months or more.

code C78) before their prescription are defined as having an aggressive disease. Patients with a waiting time above the sample median are defined as having a long waiting period.

Fig 4 presents the results on patients defined as low (panel a) and high (panel b) responders. For both sub-groups, the point estimates suggest higher mortality for patients given AA. However, the difference in effects is far more distinct for the low respondents.

The results for patients with non-aggressive (panel a) and aggressive (panel b) disease are presented in Fig 5. We can see a clear and, most often, statistically significant increased mortality if patients with non-aggressive disease are prescribed AA instead of ENZ. The pattern is less clear for patients with an aggressive disease, and all estimates are statistically insignificant.

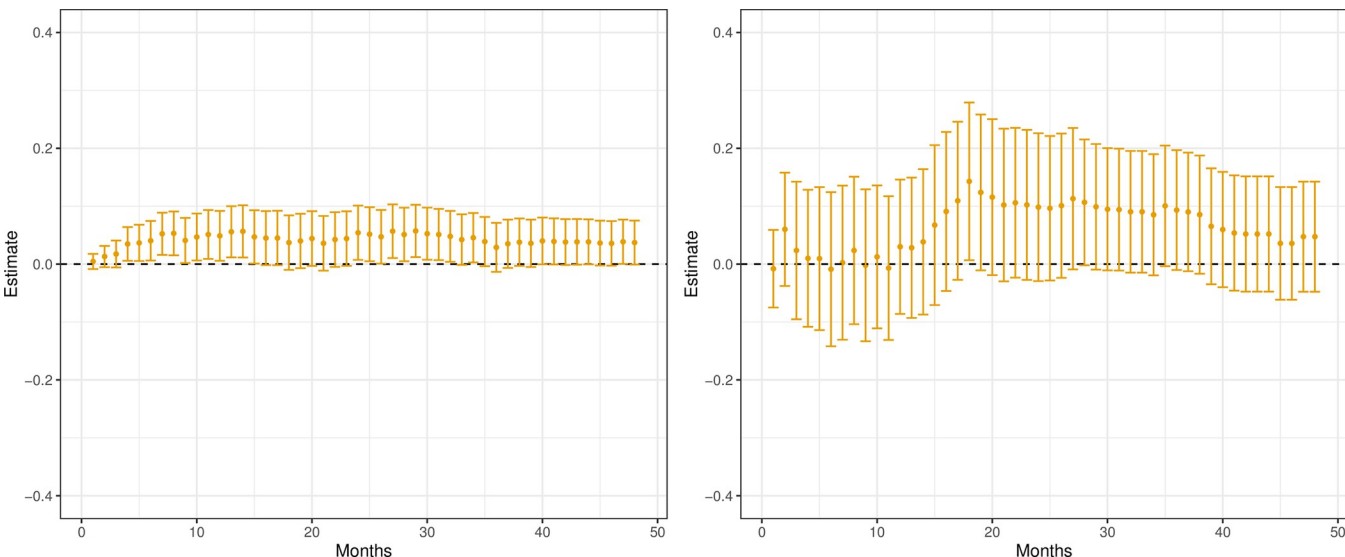

**Fig 5. CATE mortality, estimates (•) and 95% Bonferroni corrected confidence intervals.** Aggressive and non-aggressive disease. (a) Non-aggressive disease, i.e. patients without a hospital record of visceral metastases before AA or ENZ treatment. (b) Aggressive disease, i.e. patients with a hospital record of visceral metastases before AA or ENZ treatment.

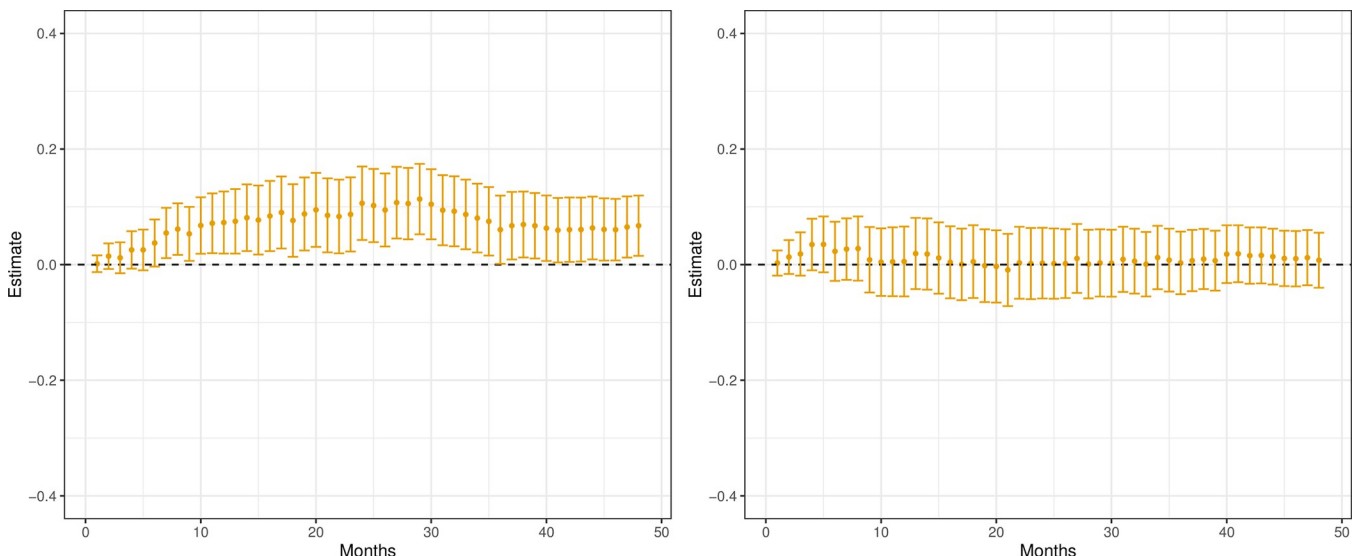

**Fig 6. CATE mortality estimates (•) and 95% Bonferroni corrected confidence intervals.** Long and short waiting times. (a) Long waiting time, i.e. patients having a longer period of time between diagnosis of prostate cancer and prescription for AA or ENZ treatment, compared to the median waiting time of 5.9 years for both drugs. (b) Short waiting time, i.e. patients having a shorter period of time between diagnosis of prostate cancer and prescription for AA or ENZ treatment, compared to the median waiting time of 5.9 years for both drugs.

The results for patients with a long (panel a) and short (panel b) waiting time to treatment are presented in Fig 6. Patients with a long waiting time for treatment display a clear reduction in mortality if given ENZ instead of AA, while no difference in effect is seen for patients with a short waiting time.

Results from the IV analysis are similar; however, there are substantially larger confidence intervals, which is why we do not present the results from these sub-analyses.

## 5.2 Skeleton-related events and severe pain

The SRE analysis results are presented in Fig 7. The Fig displays the point estimate and the 95% Bonferroni corrected confidence interval over 24 months. We find five statistically significant effects in the matched regression (months 11, 13, 14, 23 and 28).

All these effects are negative, suggesting fewer SRE from being prescribed AA than ENZ. The results from the IV analysis vary substantially over the months. However, the results should be carefully interpreted due to the small number of SRE patients.

A potential problem with this analysis is that an SRE is only observed in the data if the patient is alive. In the analysis for the matched regression sample, we excluded dead patients up to the month of the analysis. As a sensitivity analysis, we estimate the bounds of potential effects. We let all patients who die have either no morbidity outcome or a morbidity outcome (i.e. SRE = 0 or SRE = 1, respectively). If the mortality with AA is higher than with ENZ, the first case with SRE = 0 provides a lower bound estimate of the effectiveness of the NHT while the second provides an upper bound estimate on the SRE.

Panel (a) of Fig 8 presents the results from the lower bound analysis, while panel (b) presents the result from the upper bound analysis. As very few of the lower bound estimates are statistically significant and negative, while all estimates are positive and many times statistically significant, these results indicate an increase in SRE if prescribed AA rather than ENZ.

The three sub-analyses (respondents, aggressiveness, and time of prescription) on the SRE were also detailed in Johansson et al. [8]. The point estimates are, as in the main analysis,

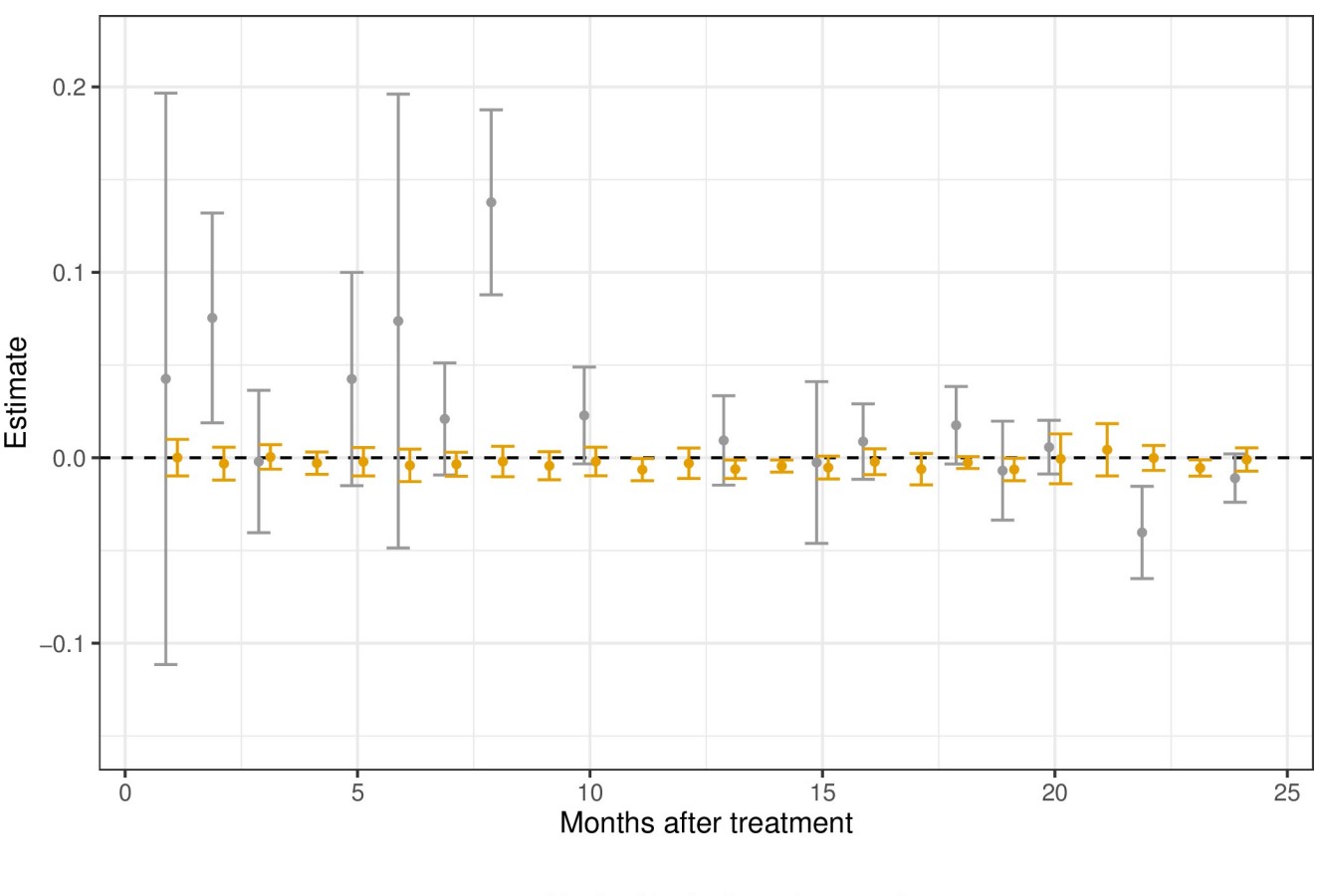

**Fig 7. CATE skeleton-related event (SRE), estimates (•) and 95% Bonferroni confidence intervals.** Bootstrap percentile intervals for the IV, with the number of replicates, R = 500.

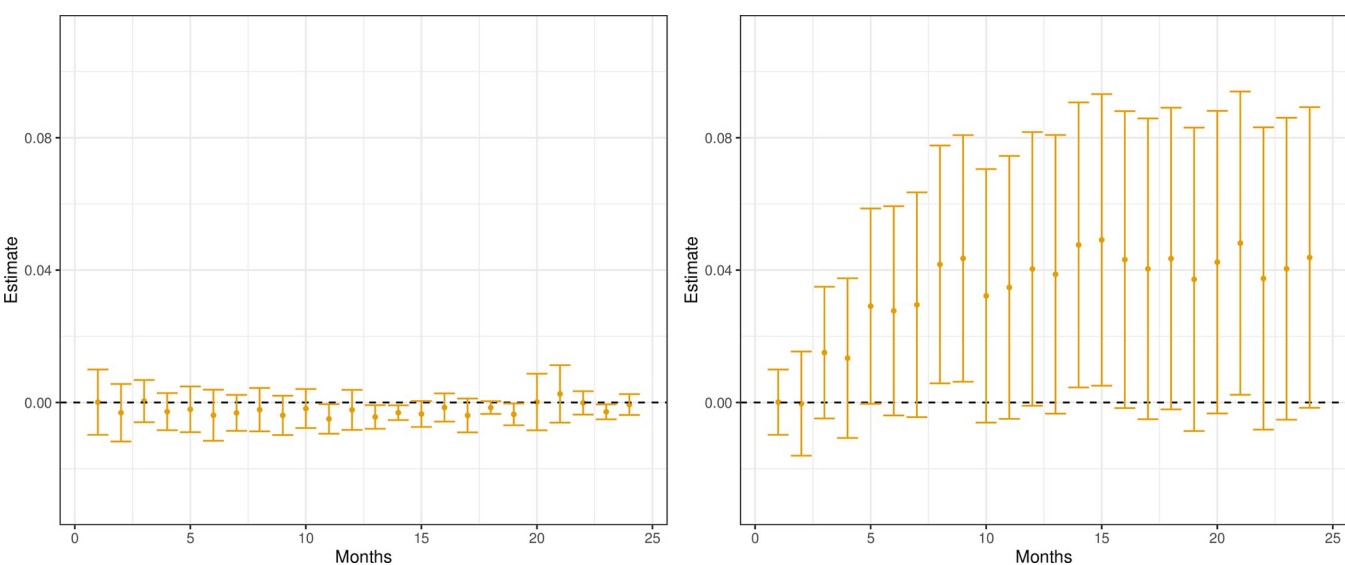

**Fig 8. CATE skeleton-related event (SRE), estimates (•) and 95% Bonferroni confidence intervals.** Lower and upper limits. (a) Lower. (b) Upper.

negative for all groups. There is some weak evidence of a more negative effect on high respondent patients (see Fig A in S3 Text) and patients with short waiting times (see Fig B in S3 Text).

As we only have 13 patients with severe pain, according to our definition of Pain, we get, as expected, no statistically significant results. However, for completeness, the results for the matched sample regression estimator are presented in S2 Fig. In the pre-analysis plan, it was stated that sub-analyses should also be made on Pain. However, as only 13 patients suffer from pain, according to our definition, sub-analyses of this outcome are not meaningful.

## 5.3 Assessing the assumptions and sensitivity analyses

At the same time as adding mortality data, data from NPCR was added to the analysis sample. These data contain more detailed information on patients' health concerning prostate cancer and allows us to assess the assumptions for the matched sample and IV analysis by estimating placebo effects on three covariates, judged by specialists as important confounders. These are PSA levels (*SPSA*), Gleason score (*GleasSa*), and metastases (*Mstad*). All covariates were measured at the time of the prostate cancer diagnosis. With three pre-measured covariates, as in the main analysis, we adjust the significance level for the individual tests using Bonferroni correction based on a five percent overall level.

The idea is that if there are statistically significant effects on these covariates when using the same regression analysis used in the main analysis, this suggests that available data from the population register are insufficient to control for confounding bias.

The NPCR does not have full coverage, and there are partially missing data on the covariates. For example, there are 170, 437 and 1,176 missing observations for *SPSA*, *GleasSa* and *Mstad*, respectively. We treat these missing data as random and remove them from the analysis. The reason is that conditional on the covariates, we found no association between missing data and treatment indicator (see S4 Text).

The results from the analysis for the matched sample regression and the IV analysis are displayed in Table 3. None of the estimated effects are statistically significant at the 1.67% level or, for that matter, the 5% level. Thus, these sensitivity analyses provide quite strong evidence of a causal effect. One drawback with the assessment is that these variables are measured at the time of diagnosis rather than at the time of prescription. Unfortunately, measurement at the time of prescription is not available in the NPCR data. However, these tests are for uncounfoundness in general, so indirectly, also for PSA at the time of prescription.

**Table 3. Results from the placebo regressions.**

|  | Estimate Standard Error | | Confidence interval | |
|---|---|---|---|---|
|  | Matched sample | | | |
| *SPSA** | -0.04 | 0.03 | [-0.1153, | 0.0425] |
| *GleasSa* | -0.07 | 0.05 | [-0.1896, | 0.0496] |
| *Mstad* | -0.04 | | [-0.0878, | 0.0078] |
|  | IV analysis | | | |
| *SPSA** | -0.10 | 0.13 | [-0.4067, | 0.2014] |
| *GleasSa* | -0.41 | 0.34 | [-1.2285, | 0.4105] |
| *Mstad* | 0.16 | 0.11 | [-0.1022, | 0.4311] |

Note: OLS cluster robust standard errors and IV bootstrap percentile method. *Standardized to have mean zero and unit variance.

**5.3.1 Rosenbaum's sensitivity analysis for hidden bias.** Rosenbaum's sensitivity analysis for hidden bias, for a matched binary outcome, is conducted for the 26 significant effects from the matched sample regression [35].

From S3 Fig, we can e.g. see that for time period 4, the upper bound for the sensitivity analysis is $\Gamma = 1.22$. This indicates that the confidence interval for the effect would include zero if an unobserved variable caused the odds ratio of treatment assignment to differ between the treatment and comparison groups, by 1.22. On average, over all 26 significant effects the upper bound $\Gamma$ is 1.14.

Results from this sensitivity analysis suggest that we should interpret the results cautiously. However, taking all things together, the results from the placebo regressions, the similar results with the IV estimator, the potential bias for the matched sample regressions must be seen as small.

## 5.4 Exploratory analysis

Johansson et al. [8] describe exploratory analyses on prostate-specific mortality and compliance with the treatment. The results from the analysis are in agreement with the results from overall mortality, but with wider confidence intervals. The results of the analysis on compliance are not statistically significant at any reasonable level of risk. For these two reasons, we defer the discussion and the presentation of the results to S5 Text.

## 6 Discussion

As a potential input to the discussion on the use of pre-published protocols in analysis of Real World Data (RWD), this paper has illustrated that an observational study based on a pre-published protocol can entail the same level of detail as a protocol for an RCT. To this end, we present the results from a comparative effectiveness evaluation of abiraterone acetate (AA) against enzalutamide (ENZ) in clinical practice, two cancer drugs prescribed to patients with advanced prostate cancer.

Based on two complementary models, we have estimated effects on all-cause mortality and two morbidity outcomes (skeleton-related events and severe pain). The designs and pre-specified analyses are described in the two pre-analysis plans [8, 9].

The results from the two analyses (matched sampling analysis and IV analysis) both show an increased mortality risk from prescribing AA compared to ENZ. These results support the findings in Tagawa et al. [25], and Schoen et al. [26]. In addition, the matched sampling analysis also suggests an increased risk of skeleton-related events.

Inference from observational studies may suffer from many forms of bias. One concerns the potential of researchers being subjective. In a model-based analysis of the outcomes, the researcher may adjust the model due to surprising results and, generally, bias the results from the analysis. This form of subjectivity bias is less of a problem with design-based studies [cf. 36] and does not exist at all as long as all relevant information can be included in a pre-analysis plan. This is the same as for an RCT.

The fact that the analysis is objective, however, does not mean that the inference is valid. Inferences from all observational analyses may be biased. One must always recognize the possibility that unobserved confounders are not balanced or, in the case of the IV design, reflect on unsubstantiated model assumptions.

The validity of the inference is assessed based primarily on the auxiliary data from the National Prostate Cancer Register. The identifying assumptions in both designs in this paper could not be rejected. In addition, as the results from the two analyses are qualitatively very similar, the findings should be of interest to the health-care profession.

## Supporting information

**S1 Text. Data delivery document.**
(DOCX)

**S2 Text. Complementary analysis.**
(DOCX)

**S3 Text. Sub-analyses of SRE.**
(DOCX)

**S4 Text. Missing data.**
(DOCX)

**S5 Text. Exploratory analysis.**
(DOCX)

**S1 Table. Included ICD codes and ATC codes.**
(DOCX)

**S2 Table. Variable descriptions (\* variable is included in factor analysis).**
(DOCX)

**S1 Fig. Per protocol analyses of mortality, estimates (•) and 5% Bonferroni corrected confidence intervals.** Excluding switchers (485 patients).
(TIF)

**S2 Fig. CATE pain, estimates (•) and 95% Bonferroni corrected confidence intervals.**
(TIFF)

**S3 Fig. Results from Rosenbaum's sensitivity analysis for hidden bias.** An unobserved confounder changing the odds ratio from one (under unconfoundedness) to Gamma imply that the confidence interval for the effects would cover zero.
(TIFF)

## Acknowledgments

The authors thank E. Grönkvist, M. Nordin, S-Å Lööf and seminar participants at Uppsala University and company representatives at Astellas Pharma and Janssen Inc for valuable comments.

## Author Contributions

**Conceptualization:** Per Johansson, Sophie Langenskiöld.

**Formal analysis:** Paulina Jonéus.

**Funding acquisition:** Sophie Langenskiöld.

**Methodology:** Per Johansson, Sophie Langenskiöld.

**Supervision:** Per Johansson, Sophie Langenskiöld.

**Writing – original draft:** Per Johansson, Paulina Jonéus, Sophie Langenskiöld.

**Writing – review & editing:** Per Johansson, Paulina Jonéus, Sophie Langenskiöld.

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
