## [Decision Letter · Decision Letter 0]

12 May 2023

PONE-D-23-06356Causal inferences and real-world evidence: A comparative effectiveness evaluation of abiraterone acetate against enzalutamidePLOS ONE

Dear Dr. Langenskiöld,

Thank you for submitting your manuscript to PLOS ONE. After careful consideration, we feel that it has merit but does not fully meet PLOS ONE’s publication criteria as it currently stands. Therefore, we invite you to submit a revised version of the manuscript that addresses the points raised during the review process.

We look forward to receiving your revised manuscript.

Kind regards,

Isaac Yi Kim, MD, PhD, MBA

Academic Editor

PLOS ONE

Reviewers' comments:

Reviewer's Responses to Questions

**Comments to the Author**

1. Is the manuscript technically sound, and do the data support the conclusions?

Reviewer #1: No

2. Has the statistical analysis been performed appropriately and rigorously? 

Reviewer #1: Yes

3. Have the authors made all data underlying the findings in their manuscript fully available?

Reviewer #1: Yes

4. Is the manuscript presented in an intelligible fashion and written in standard English?

Reviewer #1: Yes

5. Review Comments to the Author

Reviewer #1: The authors present a novel approach to studying comparative effectiveness of two medications used for the treatment of advanced prostate cancer, abiraterone and enzalutamide.

Based on the models presented, the authors estimate the effects of treatment on overall survival as well as relevant cancer outcomes (skeletal events and pain), finding higher mortality risks in abireterone versus enzalutamide. These findings appear to be in agreement with prior indirect analyses of clinical trials which have shown comparatively improved treatment responses among patients receiving enzalutamide.

A few important clarifications remain:

(1) Table 1 reveals that enzalutamide was the dominant therapy overall. Is there any data regarding regional factors influenced preferential prescription beyond subsidization?

(2) Did the authors account for prior lines of therapy? Was it possible that patients receiving AA had previously received ENZ, and were therefore receiving this medication in the setting of treatment failure?

(3) Other key variables to account for include PSA level at the time of treatment initiation, findings from bone scan (extent of disease). Were these accounted for?

Minor:

-Figure captions are provided but the figure itself is not provided

6. PLOS authors have the option to publish the peer review history of their article (what does this mean?). If published, this will include your full peer review and any attached files.

Reviewer #1: **Yes: **Michael Leapman

---

## [Author Response · Author response to Decision Letter 0]

15 Jun 2023

Dear Reviewer,

First of all, many thanks for the timely and relevant feedback we received on our paper by you and also the reviewer. 

The Reviewer responded ‘no’ to the question regarding the soundness of our paper. However, as there were no critical comments, we could not respond on this comment. We thus suspect, and hope, the reviewer mistakenly responded ‘no’ instead of ‘yes’.

In summary, we have made the following adjustments: The style requirements regarding figures, the right font sizes for the headline, the caption of the tables and figures, etc. are followed. We also clarified two relevant questions of the reviewer. 

Unfortunately, we do not have the permission to share the registry data used in the analysis, even anonymized. However, we can assist researchers wishing to replicate our results by making the data request form and all the programming code publicly available. 

However, the data request form is extensive and needs to be translated. Therefore, we cannot provide this material until we understand that you believe this to be a good idea, and our manuscript has been accepted for publication. 

Thanks to your comments, and the changes we made, we now believe that our manuscripts will be an important contribution in your distinguished journal. 

Sincerely yours

Sophie Langenskiöld

---

## [Editor Report · Decision Letter 1]

12 Jul 2023

PONE-D-23-06356R1Causal inferences and real-world evidence: A comparative effectiveness evaluation of abiraterone acetate against enzalutamidePLOS ONE

Dear Dr. Langenskiöld,

Thank you for submitting your manuscript to PLOS ONE. After careful consideration, we feel that it has merit but does not fully meet PLOS ONE’s publication criteria as it currently stands. Therefore, we invite you to submit a revised version of the manuscript that addresses the points raised during the review process.

We look forward to receiving your revised manuscript.

Kind regards,

Isaac Yi Kim, MD, PhD, MBA

Academic Editor

PLOS ONE

Additional Editor Comments:

None of the concerns raised by the reviewer has been addressed. Pls read the initial response carefully.

---

## [Author Response · Author response to Decision Letter 1]

13 Aug 2023

Dear Reviewer,

We are grateful for your help to understand what changes were missing in our resubmitted version, and apologize for not having addressed all comments for item 5. 

We have now addressed all comments also for item 5 both in the file responding to your suggestions for change (addressed comments), and also in the manuscript. We also clarify on the file in question on which row/s in the clean manuscript all changes are made for item 5, and also for the other items. Thereby, we hope to alleviate for you to understand what changes are made as a result of your review of our manuscript. 

We also will take the opportunity to thank you for your relevant comments. They have for sure clarified our paper. 

Kind regards,

Sophie Langenskiöld

Per Johansson

Paulina Joneus

---

## [Editor Report · Decision Letter 2]

6 Sep 2023

PONE-D-23-06356R2Causal inferences and real-world evidence: A comparative effectiveness evaluation of abiraterone acetate against enzalutamidePLOS ONE

Dear Dr. Langenskiöld,

Thank you for submitting your manuscript to PLOS ONE. After careful consideration, we feel that it has merit but does not fully meet PLOS ONE’s publication criteria as it currently stands. Therefore, we invite you to submit a revised version of the manuscript that addresses the points raised during the review process.

I would like to emphasize that the revision must contain a separate analysis without the 10% of men who were exposed to both abiraterone and enzalutamide. 

We look forward to receiving your revised manuscript.

Kind regards,

Isaac Yi Kim, MD, PhD, MBA

Academic Editor

PLOS ONE

Additional Editor Comments:

This manuscript's objective is to demonstrate a new methodology in comparing outcomes between abiraterone acetate and enzalutamide. Yet, about 10% of patients were exposed to both drugs. The authors argue that such contamination is acceptable because the study is an intention to treat study. Nevertheless, given the contamination rate, the authors must demonstrate that the differences mortality holds true even after the removal of the contaminated group.

In short, a separate analysis between the two groups must be carried out after removing the 10% contaminated group.

---

## [Author Response · Author response to Decision Letter 2]

8 Sep 2023

Dear Reviewer,

We are very grateful for the time you have taken to review our paper. Your last suggestion of also conducting an as-protocol analysis was highly relevant. It is now included in our manuscript, and we believe it strenghens our results.

Kind regard,

Sophie Langenskiöld

Per JOhansson

Paulina Jonéus

---

## [Editor Report · Decision Letter 3]

12 Sep 2023

PONE-D-23-06356R3Causal inferences and real-world evidence: A comparative effectiveness evaluation of abiraterone acetate against enzalutamidePLOS ONE

Dear Dr. Langenskiöld,

Thank you for submitting your manuscript to PLOS ONE. After careful consideration, we feel that it has merit but does not fully meet PLOS ONE’s publication criteria as it currently stands. Therefore, we invite you to submit a revised version of the manuscript that addresses the points raised during the review process.

Please provide more details for figure legends. This especially true for the supplementary figures.

We look forward to receiving your revised manuscript.

Kind regards,

Isaac Yi Kim, MD, PhD, MBA

Academic Editor

PLOS ONE
---

## [Author Response · Author response to Decision Letter 3]

13 Sep 2023

Dear,

Again, many thanks for valuable comments!

Kind regards,

Sophie Langenskiöld

Per Johansson

Paulina Jonéus

---

## [Editor Report · Decision Letter 4]

4 Oct 2023

Causal inferences and real-world evidence: A comparative effectiveness evaluation of abiraterone acetate against enzalutamide

PONE-D-23-06356R4

Dear Dr. Langenskiöld,

We’re pleased to inform you that your manuscript has been judged scientifically suitable for publication and will be formally accepted for publication once it meets all outstanding technical requirements.

Kind regards,

Isaac Yi Kim, MD, PhD, MBA

Academic Editor

PLOS ONE
---

## [Editor Report · Acceptance letter]

16 Oct 2023

PONE-D-23-06356R4 

Causal inferences and real-world evidence: A comparative effectiveness evaluation of abiraterone acetate against enzalutamide 

Dear Dr. Langenskiöld:

I'm pleased to inform you that your manuscript has been deemed suitable for publication in PLOS ONE. Congratulations! Your manuscript is now with our production department. 

Kind regards, 

on behalf of

Dr. Isaac Yi Kim 

Academic Editor

PLOS ONE